# SWE-Bench+: Enhanced Coding Benchmark for LLMs

## Abstract

Large Language Models (LLMs) in Software Engineering (SE) can offer assistance for coding. To facilitate a rigorous evaluation of LLMs in practical coding contexts, Carlos et al. introduced the *SWE-bench* dataset, which comprises 2,294 real-world GitHub issues and their corresponding pull requests, collected from 12 widely used Python repositories. Several impressive LLM-based toolkits recently are developed and evaluated on this dataset. However, a systematic evaluation of the quality of SWE-bench remains missing. In this paper, we addressed this gap by presenting an empirical analysis of the *SWE-bench* dataset. We conducted a manual screening of instances where *SWE-Agent + GPT-4* successfully resolved issues by comparing the model-generated patches with the actual pull requests. SWE-Agent+GPT-4 was at the top of SWE-bench leaderboard during the time of our study. Our analysis reveals some critical issues with the *SWE-bench* dataset: 1) 32.67% of the successful patches involve "cheating" as the solutions were directly provided in the issue report or the comments. We refer to as 'solution leakage' problem. 2) 31.08% of the passed patches are suspicious patches due to weak test cases, i.e., the tests were not adequate to verify the correctness of a patch. When we filtered out these problematic issues, the resolution rate of SWE-Agent+GPT-4 drops from 12.47% to 3.97%. We also observed that the same data qualify issues also exist in the two variants of SWE-bench, i.e., *SWE-bench Lite* and *SWE-Bench Verified*. In addition, over 94% of the issues were created before LLM's knowledge cutoff dates, posing potential data leakage issues.

The critical problem in the current versions of *SWE-bench* dataset motivated us to refine it to build a more rigorous evaluation dataset *SWE-Bench+*. We created SWE-bench+ by collecting GitHub issues that were created after the training cutoff dates of the LLMs to prevent the potential data leakage problem. We also ensure that the issues collected do not contain solutions in their reports or comments. After carefully analyzing the passed instances from the *SWE-Agent + GPT-4* model with the new dataset, *SWE-Bench+*, we observed a decline in the pass rate, dropping from 3.97% (as seen on the refined *SWE-Bench*) to a resolution rate of 0.55%. We further evaluated *SWE-RAG + GPT-4*, *SWE-RAG + GPT-3.5*, and *AutoCodeRover + GPT-4o* models on the new dataset to verify our findings, where the resolution rates of the models drop significantly, which are 0.73%, 0.55%, and 3.83%, respectively.

## 1 Introduction

*SWE-bench* aka Software Engineering Benchmark dataset is created to systematically evaluate the capabilities of an LLM in resolving software issues. The dataset contains 2,294 complex issues from GitHub Jimenez et al. (2024). Given as input the issue information to an LLM, the task for the LLM is to modify the code base to address the issue (i.e., resolution). Each input for an issue consists of a description and a pull request with a reference to the corresponding buggy code repository. Each issue can be either a bug report or a new feature request. The pull request contains the code changes made by developers to address the issue, along with test cases designed to check if the feature is properly implemented or if the bug is successfully fixed. Two variants of the *SWE-bench* datasets are recently developed: *SWE-bench Lite*[1] and *SWE-bench Verified*[2]. *SWE-bench Lite* focuses on 300 issues related to bug fixing. *SWE-bench Verified* contains 500 verified issues with clear issue descriptions and strong test cases.

A significant body of work from both academia and industry has so far utilized *SWE-bench* and its variants to develop and to test LLM coding capabilities Chen et al. (2024); Zhang et al. (2024a); Xia et al. (2024); Yang et al. (2024b); Zhang et al. (2024c); Rosa et al. (2024); Zan et al. (2024). Given an issue and its associated buggy code repository, these LLM-based approaches can perform a series of complex tasks, such as reasoning about the target bug's location, analyzing the root cause of the issue, proposing strategies for fixing the bug, and ultimately writing a patch to fix the

---

[1] https://www.swebench.com/lite.html
[2] https://openai.com/index/introducing-swe-bench-verified/

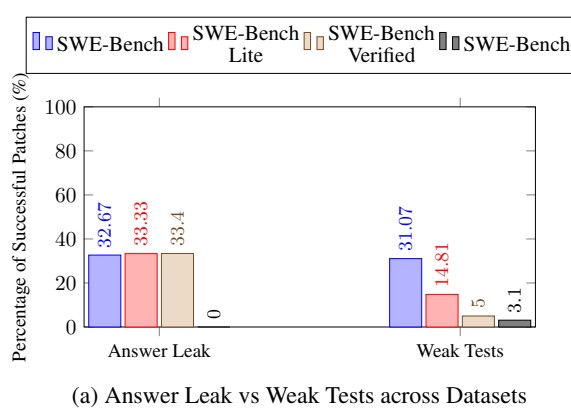 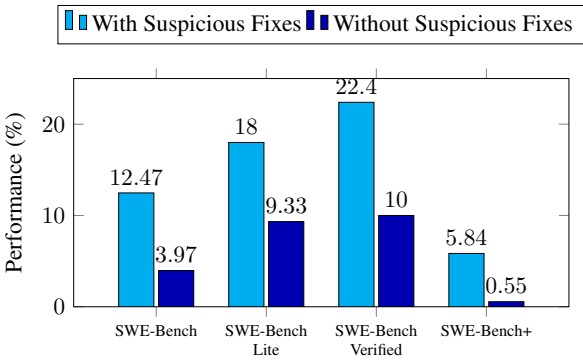

(a) Answer Leak vs Weak Tests across Datasets     (b) Performance of SWE-Agent + GPT-4 across datasets.

Figure 1: Comparison of performance metrics and patterns across *SWE-bench* datasets

issue. Within less than one year, the resolution rate on *SWE-bench Full* increased from 0.17% (for RAG+GPT3.5) to around 22.00% (for Honeycomb). The performance of the LLMs on *SWE-bench Lite* and *Verified* went up to 45%.

However, *are the LLMs actually resolving the issues in SWE-bench?*

In this paper, we answer the above question by offering two contributions. **First,** we present an empirical study of state-of-the-art (SOTA) LLMs on SWE-bench Full that explores 1) the quality of SWE-bench issues with a focus on the testing adequacy of the test cases used for validating patches and 2) the quality of patches generated by the LLMs to fix the issues. **Second,** we present an enhancement of *SWE-bench Full*, which we call *SWE-bench+*.

During the time of our empirical study, *SWE-Agent+GPT-4* was at the top of the *SWE-bench* online leaderboard. Other top approaches (e.g., Honeycomb, Amazon Q Developer Agent, and Factory Code Droid) were either closed-sourced commercial tools or not verified by the SWE-bench team regarding reproducibility. SWE-agent Yang et al. (2024b) allows LLM agents to execute basic file operations via shell commands to achieve interaction between the LLM engine and a software repository. First, we picked issues that were claimed as resolved by *SWE-Agent+GPT-4*. We did this by filtering only the instances with evaluation logs showing that all tests passed. Second, we performed a patch validation study by comparing the gold patches (i.e., original) to the model patches (i.e., generated). We did this by comparing the files changed, the lines changed, and the code changes made in the fixes (both original and generated). Third, we determined eight patterns in the generated fixes by reviewing the issue reports, the corresponding tests, and the available discussions of issues (which are treated as hints to LLMs).

Our identified six patterns in the 251 SWE-Agent+GPT-4 patches can be broadly divided into two types: *suspicious fixes* and *correct fixes*. Suspicious fixes corresponded to 63.75% (i.e., 160) of the patches. Two patterns were prevalent in those fixes: 1) **Answer Leak.** In 32.67% of the resolved instances, the solutions were outlined directly in the issue reports or comments. 2) **Weak Tests.** In 31.08% of the resolved instances, the changes made by the model are either incorrect, incomplete, or applied to different files or functions compared to the gold patch. Despite these discrepancies, the changes pass the tests, indicating that the tests are too weak to catch such errors. In addition, the dataset can also suffer from potential data leak issues. This is because 94% of the instances in SWE-bench and their pull requests were created prior to the training cut-off dates of the LLMs, meaning that all issue reports in the full SWE-bench dataset may have been exposed to the LLMs during their training phases, raising concerns about potential data leakage.

Based on the above observation, we considered fixes corresponding to issues with answer leakage and weak tests as 'suspicious fixes'. In Figure 1a, we show the distribution of such fixes in the three literature datasets (i.e, SWE-bench full, lite, verified). In Figure 1b, we show that after filtering these suspicious fixes, the correct resolution rate of *SWE-Agent+GPT-4* dropped to 3.97% from 12.47%. This drastic drop in resolution rate raises concerns about the robustness of the model-generated patches and the reliability of the SWE-bench dataset itself.

To address the problems in *SWE-bnech* datasets, we created *SWE-bench+* dataset, which ensures that: 1) the data were created after the models' training cut-off dates, and 2) the issues do not include solutions in the issue description or comments. *SWE-bench+* dataset is created by following the same data collection methodology described in the *SWE-Bench* dataset; except we filtered out issues with answer leakage problems. Considering the training cut-off dates of the LLMs used in our study—GPT-3.5 (turbo-16k-0613) with a cut-off in September 2021, GPT-4 (1106) up to April 2023, and GPT-4o (2024-05-13) up to October 2023, we opted to collect data starting a month after the most recent model's cut-off date. To the end, we gathered issues from the period of 2023-11-01 to 2024-08-22. As we show

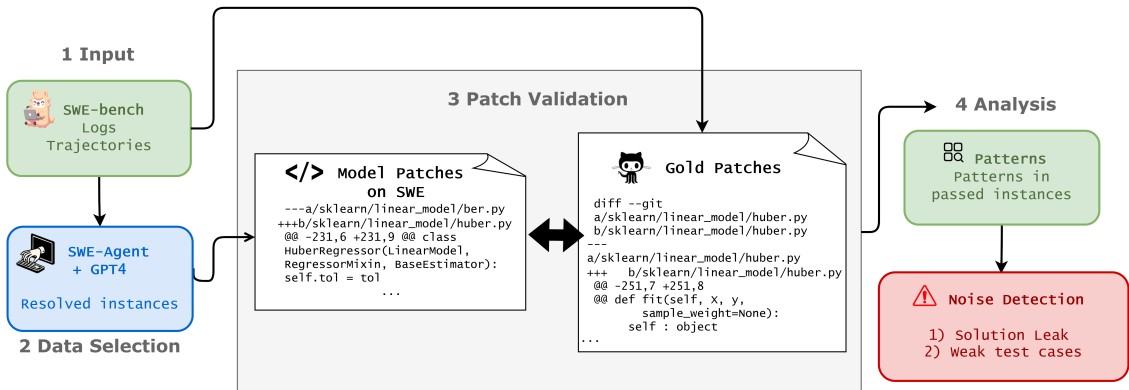

Figure 2: Overview of robustness analysis for *SWE-Bench* datasets

in Figure 1a, *SWE-bench+* has no issues with solution leakage whereas all the other three datasets suffer from this. *SWE-bench+* also has the lowest proportion of issues with weak test cases among all the SWE-bench variants. As such, we consider SWE-bench+ as the most robust dataset among the available *SWE-bench* variants. When we ran *SWE-Agent+GPT-4* on *SWE-bench+* dataset, its resolution rate dropped to 0.55% (see Figure 1).

We further evaluated the *SWE-RAG + GPT-4*, *SWE-RAG + GPT-3.5*, and *AutoCodeRover + GPT-4o* models on the new dataset to verify our findings. The resolution rates of the models dropped significantly, with the new rates being 0.73%, 0.55%, and 3.83%, respectively. In comparison, the previously reported resolution rates on the SWE-Bench leaderboard were 1.31% for *SWE-RAG + GPT-4*, 0.17% for *SWE-RAG + GPT-3.5*, and 18.83% for *AutoCodeRover + GPT-4o*.

**Artifacts.** While working on merging our *SWE-bench+* to *SWE-bench* project repository, we release the dataset of *SWE-bench+* to help other researchers replicate and extend our study[3].

## 2 ROBUSTNESS ANALYSIS OF SWE-BENCH

We conducted an empirical study of *SWE-Agent+GPT-4* generated patches for issues in the *SWE-bench Full* dataset. The goal of our study was to identify whether the patches exhibit any potential problems. Figure 2 outlines the major steps we followed in our study. The input is the set of all issues in *SWE-bench*. Each issue contains a description and the patch to address the issue. Each patch is a diff of code changes. We call this a "gold patch". We picked *SWE-Agent + GPT-4* and applied it on each issue to create a fix. We refer to the model output for an issue as a "generated patch". We then compared the gold and generated patches to an issue by analyzing the corresponding files changed in the pull requests on GitHub with the same *instance_id*. As we studied the model generated patches, we also examined the logs and trajectories generated by the model. Logs provide the step-by-step execution of the models. The trajectory data provide a detailed record of the models' decision-making processes while making a resolution as a patch.

To reduce potential biases during the comparison between gold and generated patches, three authors independently performed the patch validation study. Each author carefully examined the files and lines changed, reviewed the issue descriptions, and evaluated the implementation styles and intentions behind both the model-generated and developer-generated patches. The disagreements were resolved through a broader discussion involving all the authors.

For the model generated patches, we focused on instances where the generated patches resolved the issue and passed all associated tests. As a result, we identified 251 instances from the *SWE-Bench Full* dataset. Note that, to ensure the patches passed all tests, we reviewed the evaluation logs of 286 instances initially marked as resolved by *SWE-Agent + GPT-4* in the *results.json* file from the *SWE-Bench Full* evaluation repository and selected only those with logs confirming that all tests passed following the application of the generated patches.

### 2.1 CRITICAL ISSUES OF *SWE-Bench*

Among the 251 generated patches that passed all test cases in *SWE-bench Full* dataset, we found several patches as problematic/suspicious. Table 1 outlines six patterns in the 251 generated patches, four related to the suspicious fixes

---

[3]https://zenodo.org/records/13879453

Table 1: Patterns found among the 251 successful patches generated by *SWE-Agent + GPT-4*

| Type | Pattern | Numbers (percentage) | Root cause |
|------|---------|----------------------|------------|
| Suspicious fixes | Solution leak | 82 (32.67%) | solution leakage |
| | Incorrect fixes | 32 (12.75%) | weak tests |
| | Different files/functions changed | 9 (3.59%) | weak tests |
| | Incomplete fixes | 37 (14.74%) | weak tests |
| Correct fixes | Different fixes from gold patches | 76 (30.27%) | – |
| | More comprehensive fixes than gold patches | 15 (5.98%) | – |

and two related to the correct fixes. To explain each pattern, we provide definitions, including the number of instances associated with each pattern and the likely root causes. We discuss each pattern below.

### 2.1.1 PATTERNS EXTRACTED FROM THE SUSPICIOUS FIXES

We observed four patterns in the suspicious fixes, one is attributed to solution leakage and the other three are attributed to weak test case problems in the *SWE-bench* dataset.

**1. Solution leak:** represents instances where the solution to the issue is clearly outlined in the issue description or comments on GitHub. Since both the issue descriptions and comments (referred to as *hints_text* in the SWE-Bench study) are provided as input to the models, these LLM models can extract the solutions directly from this information instead of generating it independently. 32.67% of the successfully resolved issues followed this pattern, making it the most common among resolved patches. This raises significant concerns about a model's actual performance and the validity of the SWE-Bench instances as benchmarks. If a model is simply copying the solution it already has access to, it isn't demonstrating true problem-solving capabilities but rather replicating what is provided, thus limiting the assessment of its ability to generate new solutions. The example shown in Figure 3 illustrates issue report 16669[4] from the *sympy* project, where the issue description provided the exact solution code patch required to resolve the issue, which makes it possible for the model to directly copy the solution from the issue report and generate the same solution as provided.

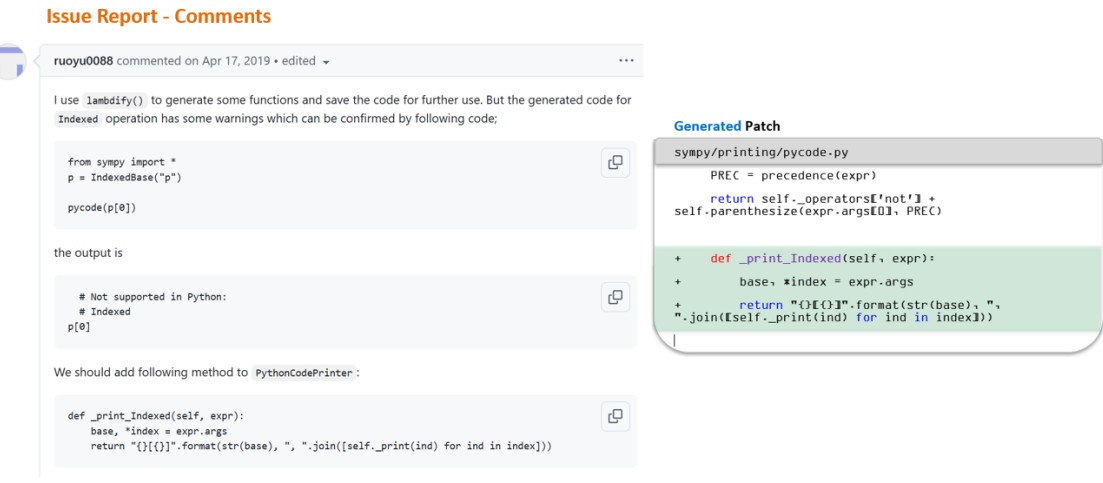

Figure 3: Solution Leakage in issue report for sympy-16669

**2. Incorrect fixes:** refer to cases where the model-generated patches provide incorrect solutions, yet pass the test cases when they should have failed. This pattern was present 12.75% of the passed instances. Suggesting a weakness in test cases where the functionality of the issue resolution is not correctly captured. The fact that incorrect patches

---
[4]https://github.com/sympy/sympy/issues/16669

can pass the test cases raises suspicion about the relevance and accuracy of the test cases in assessing whether the issue has been fully resolved. Figure 4 shows a comparison between the model-generated patch and the gold patch for django-32517[5]. According to the issue description, a new functionality is needed to reverse a Python *OrderedSet* by implementing the *__reversed__* function. The gold patch demonstrates the correct behavior, where the entire dictionary is reversed, while the generated patch only reverses the dictionary's keys. As a result, the two patches produce entirely different outputs, as they apply different methods to the dictionary.

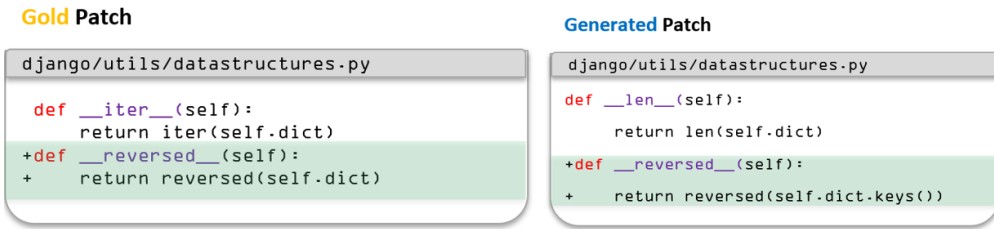

Figure 4: Incorrect fix generated by the model for django-32517

**3. Different files/functions changed:** This pattern refers to cases where the model-generated patches modify files or functions unrelated to the issue at hand. These files differ from those altered in the gold patch, yet the model's patches still pass the test cases despite this discrepancy. This highlights a weakness in the model's ability to accurately locate and address the source of the issue. The fact that the test cases pass, even though changes were made in irrelevant files, suggests that the test cases are either weak or irrelevant and should have failed in detecting the incorrect modifications. Figure 5 presents an example from issue-26093 of Matplotlib project[6], where the model-generated patch modifies the *cbook.py* file, while the gold patch makes changes to the *_axes.py* file. This shows that the model's patch affects a completely different file from the gold patch, highlighting the model's inability to accurately identify the correct file containing the bug.

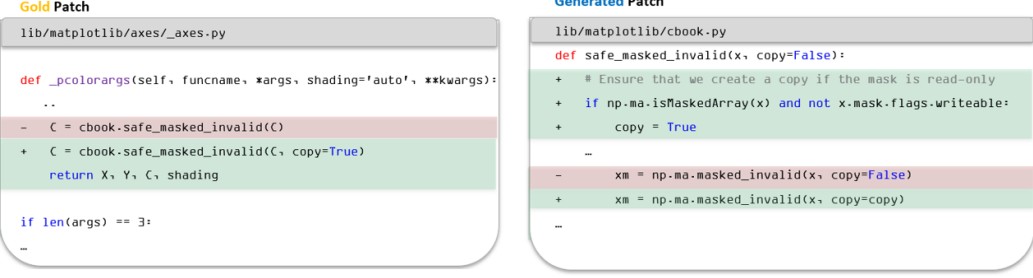

Figure 5: Different files changed by model for issue-26093 of Matplotlib

**4. Incomplete fixes:** This pattern refers to model-generated patches that offer incomplete implementations compared to the gold patches, often omitting critical details. For instance, some patches include only partial if-else statements, neglecting edge cases that the gold patch addresses. Although the model-generated patches follow the correct implementation approach, they overlook important aspects that could lead to failures in production or when handling edge cases. This underscores a weakness in the test cases, as they fail to capture the finer details necessary for a comprehensive issue resolution.

The example provided in Figure 6 shows the same change being made by the model and the one made by the developers in the gold patch[7]. The gold patch provides a complete fix while the model patch provides a partial fix. Specifically, the gold patch properly handles the detection of an event loop in the current thread by including a *try-except* block to catch *RuntimeError* when an event loop is unavailable and checks if the event loop is running before raising an exception. Additionally, it wraps the entire logic in a condition that checks the environment variable *DJANGO_ALLOW_ASYNC_UNSAFE*. In contrast, the generated patch is missing critical parts of this logic, such as the *try-except* block and the check for a running event loop. As a result, the model-generated patch is incomplete, missing key error handling and flow control that are necessary for ensuring safe operation.

---

[5]https://code.djangoproject.com/ticket/32517

[6]https://github.com/matplotlib/matplotlib/issues/26093

[7]https://code.djangoproject.com/ticket/31056

Figure 6: Incomplete fix generated by the model for django-31056

### 2.1.2 Patterns extracted from the correct fixes

**1. Different fixes from gold patches:** This pattern refers to cases where the model-generated patches present an entirely different solution to the issue compared to the gold patch. Although the coding style and implementation differ, the model-generated patches correctly resolve the issue. Instances in this pattern are considered a correct resolution of the issues.

**2. More comprehensive fixes:** In contrast to the previous pattern, this refers to instances where the model-generated patches are more comprehensive than the gold patches. For example, the model-generated patches may include additional if-else cases or other logic that the gold patch does not address. This showcases a strength of LLMs, as they can generate more thorough solutions that cover scenarios developers might overlook, resulting in potentially safer and more robust solutions (e.g., by incorporating try-catch statements or handling edge cases more effectively). We consider this pattern as a correct resolution of the issue.

### 2.2 Updated Resolution Rate of SWE-Agent + GPT-4 on SWE-Bench Full

After identifying these patterns, we recalculated the resolution rate of *SWE-Agent + GPT-4*, focusing exclusively on patches classified under the correct fixes patterns. This new resolution criteria resulted in a significant drop in the resolution percentage, as shown in Figure 1b. The performance of *SWE-Agent+GPT-4* decreased from 12.47% to 5.49% when considering only correct fixes—those where the model generated either correct but different implementations from the gold patch (30.27% of the time) or more comprehensive fixes than the gold patch (5.98% of the time). We excluded all suspicious fixes, which included instances where the model generated incorrect solutions (12.75%), where the issue report contained a direct solution (32.67%), where changes were made in files unrelated to the gold patch (3.59%), and cases where the model produced incomplete fixes, missing critical details of the solution (14.74%).

### 2.3 Updated Resolution Rate of SWE-Agent+GPT-4 on SWE-bench Lite and Verified

Two new variants of *SWE-Bench* are recently developed, *SWE-Bench Lite* and *SWE-Bench Verified*, each designed with different goals. *SWE-Bench Lite* focuses on instances with lower evaluation costs and increased accessibility. On the other hand, *SWE-Bench Verified* aims to provide a curated subset of *SWE-Bench*, where human annotators filter out issues with underspecified descriptions or weak unit tests that might reject valid solutions. However, neither of these new datasets addresses the solution leakage problem, which was the primary motivation for our study. To investigate this, three of the authors reviewed all issue reports for the instances in *SWE-Bench Lite* and *SWE-Bench Verified* for cases of solution leakage. *In SWE-Bench Lite*, we identified 18 instances where the solution was directly provided in the issue description or discussion on GitHub. Similarly, in *SWE-Bench Verified*, 37 instances contained direct solutions in either the issue description or the discussion on GitHub. Table 2 shows more details.

Table 2: Patterns found among SWE-Bench Lite (total passed: 54, 18.0%) and SWE-Bench Verified (total passed: 112, 22.4%) datasets with successful patches generated by SWE-Agent + GPT-4.

| Pattern | SWE-Bench Lite | Percentage (Lite) | SWE-Bench Verified | Percentage (Verified) |
|---|---|---|---|---|
| **Incorrect fixes** | 5 | 9.26% | 14 | 12.50% |
| **Incomplete fixes** | 3 | 5.56% | 11 | 9.82% |
| **Different Files/Functions Changed** | 0 | 0.00% | 0 | 0.00% |
| **Solution Leak** | 18 | 33.33% | 37 | 33.04% |

We also observed additional suspicious fixes in both *SWE-Bench Lite* and *SWE-Bench Verified*, as summarized in Table 2. Specifically, 9.26% of fixes in *SWE-Bench Lite* and 12.50% in *SWE-Bench Verified* were incorrect fixes generated by *SWE-Agent+GPT-4*. Additionally, 5.56% of fixes in *SWE-Bench Lite* and 9.82% in *SWE-Bench Verified* were incomplete fixes from the same model. Notably, there were no issues involving changes to different functions or files, as the model correctly identified the buggy file in all cases for both datasets. Despite this, the overall suspicious fix patterns led to 48.14% suspicious fixes in *SWE-Bench Lite* and 55.36% in *SWE-Bench Verified*, significantly reducing the resolution rates—from 18% to 9.33% in *SWE-Bench Lite* and from 22.4% to 10.0% in *SWE-Bench Verified*.

## 3 BUILDING SWE-BENCH+

To address the issues of the current *SWE-Bench* datasets and ensure a more accurate evaluation of the models' effectiveness in resolving issues, we utilized a new dataset, *SWE-Bench+*, which focuses on issues with no clear solution provided in the issue report and without potential risk of data leakage. The primary objective of *SWE-Bench+* is to assess the models' ability to generate accurate patches for real-world GitHub issues without the risk of bias or prior exposure to the solution. To maintain consistency and fairness when comparing the resolution rates of the models on*SWE-Bench* and *SWE-Bench+*, we followed the same data collection methodology outlined in the SWE-Bench study, using their open-source scripts.

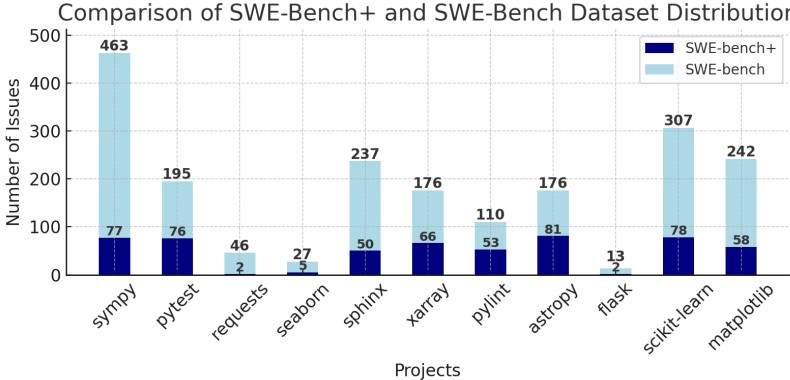

Figure 7: SWE-Bench+ dataset compared to SWE-Bench

First, we selected the same 12 projects from *SWE-Bench*, except Django, which was excluded as its issues are now tracked outside of GitHub. Given that the LLMs we used in the models (GPT-4, GPT-3.5, and GPT-4o) were all trained on data up to October 2023, we collected issues that appeared after October 2023. Once the issues were collected, we applied the same filtering process described in the *SWE-Bench* study. This included using attribute filtering to retain only issues that resolve a problem and contribute tests, followed by an execution filter to keep only issues that install successfully and their PRs pass all tests. We then manually check all the instances to eliminate all the instances with a clear solution details in the issue report. By following this method, we obtained 548 task instances (issues) from the selected projects, with the distribution shown in Figure 7.

Table 3: Performance of different models on *SWE-bench+*

| Pattern | SWE-RAG+GPT-4 | SWE-RAG+GPT-3.5 | SWE-Agent+GPT-4 | AutoCodeRover+GPT-4o |
|---|---|---|---|---|
| Correct | **4** | 3 | 3 | **21** |
| (Suspicious) Different files/functions changed | **16** | 11 | 14 | 7 |
| (Suspicious) Incorrect fixes | 2 | 6 | 3 | **15** |
| (Suspicious) Incomplete fixes | 1 | 0 | 0 | **3** |
| Total | 23 | 20 | 32 | **42** |

## 4 ROBUSTNESS OF SWE-BENCH+

After collecting the data and identifying the correct repository versions, the next step was to run the selected models to generate patches that would address the issues. The models chosen for this task were *SWE-RAG* with GPT-3.5 (turbo-16k-0613), *SWE-RAG* with GPT-4 (1106), *SWE-Agent* with GPT-4 (1106), and *AutoCodeRover* (v20240620) paired with GPT-4o (2024-05-13). These models were selected based on their performance, large context window, cost, and the fact that they are open-source, making them more favorable compared to other options.

At the beginning of the project, *SWE-Agent* had the highest resolution rate on the SWE-Bench leaderboard among all open-source models. Since we studied this model thoroughly in relation to SWE-Bench, we selected it to ensure a fair comparison. *SWE-RAG+GPT-4* and *SWE-RAG+GPT-3.5* were also chosen for the study due to their higher token limits compared to the *Claude* models. Initial trials with *SWE-Agent+Claude 3 Opus* and RAG-based Claude models (*SWE-RAG+Claude 3 Opus* and *SWE-RAG+Claude 2*) revealed that the token limits were exceeded before completing experiments on all 548 SWE-Bench+ instances. As the leaderboard frequently updates, with new models surpassing older ones, *AutoCodeRover+GPT-4o* recently ranked among the top three models, and since it is open-source, we also included it in the study. We following their instructions to run the models on our *SWE-Bench+* dataset.

Our evaluation followed a four-step technique, described in Figure 2, to ensure that the resolution rates of the models on SWE-Bench+ accurately reflected their correctness. Specifically, in **Step 1**, we stored all the model-generated diff files as patches in separate *json* files, we then utilized the scripts provided by the SWE-Bench to evaluate the generated patches. In **Step 2**, once the evaluation results were generated, we manually reviewed the instances marked as *resolved* and selected only those where all tests in *PASS_TO_PASS* and *FAIL_TO_PASS* had passed. In other words, we filtered out instances where the patches were successfully applied, and all associated tests passed. In **Step 3**, after filtering the resolved instances where all tests passed, we analyzed the resolution rates of the models. Finally, in **Step 4**, we performed a similar patch validation study as described in Section 2, comparing the model-generated patches to the gold patches. We found that solution-leak related issues are now resolved. However, a prominent issue with weak test cases persists. As shown in Table 3, on average, around 67.72% of the resolved instances did not truly resolve the issue, despite passing all the tests. The prominent pattern identified was the models' inability to accurately locate the buggy files or lines. This raises concerns about the models' ability to locate the bug. In other cases, the model generated incomplete or incorrect fixes, not resolving the bug. These findings show that the issue with weak tests still persists and needs future investigation.

As a result of the patch validation analysis, the resolution rates were 0.73% for *SWE-RAG+GPT-4*, 0.55% for *SWE-RAG+GPT-3.5*, 0.55% for *SWE-Agent+GPT-4*, and 3.83% for *AutoCodeRover+GPT-4o*. These rates are significantly lower than the reported resolution rates on the *SWE-Bench* leaderboard were 1.31% for *SWE-RAG + GPT-4*, 0.17% for *SWE-RAG + GPT-3.5*, and 18.83% for *AutoCodeRover + GPT-4o*. This drop in performance for all models when moving from *SWE-Bench Full* to *SWE-Bench+*, highlights the impact of excluding suspicious fixes on the overall resolution rates.

## 5 EFFECTIVENESS-AWARE EVALUATION

Despite the notable success in resolving issues, we also observed significant variations in the costs of the approaches tested. While some models excelled in accuracy and efficiency, they required more computational resources, longer processing times, and higher costs. Specifically, *SWE-Agent+GPT-4* and *AutoCodeRover+GPT-4* had the longest code generation times, with *SWE-Agent+GPT-4* averaging around 4 minutes per instance, resulting in a total of approximately 37 hours to generate patches for SWE-Bench+ issues. The average generation time for *AutoCodeRover* was 4.5 minutes per instance, resulting in 41 hours in total to generate patches for all SWE-Bench+ instances, making it a top performer in issue resolution for SWE-Bench+ instances. However, this disparity highlights the trade-offs

Table 4: Average cost of different models on *SWE-Bench+*

| Model | Avg cost per instance | cost per issue fixing | Avg time per instance |
|---|---|---|---|
| *RAG+GPT 4* | $0.24 | $32.5 | 30 seconds |
| *RAG+GPT 3.5* | $0.05 | $10.0 | 30 seconds |
| *SWE_AGENT+GPT 4* | $3.59 | $655.0 | 4 minutes |
| *AutoCode Rover+GPT 4o* | $0.46 | $12.61 | 4.5 minutes |

between performance and cost-effectiveness, particularly for models like *SWE-Agent+*, where balancing time, cost, and resource allocation is critical for real-world applications.

In terms of cost, we identified two metrics, i.e., the average cost per instance (calculated by dividing the total cost by the 548 instances in SWE-Bench+ that the models were tested on) and the effectiveness-aware cost per instance (calculated by dividing the total cost by the number of instances successfully resolved by the model). The detailed cost of each model measured by the two metrics is shown in Table 4. Overall, *SWE-Agent+GPT-4* was the most expensive model, with an average cost of $0.24 per instance and an effectiveness-aware cost of $32.5 per issue fixed, as shown in Table 4. Despite its high cost, its performance was comparable to *RAG+GPT-4*, which was much more cost-efficient, with an average cost of $0.05 per instance and an effectiveness-aware cost of $10.0. On the other hand, *AutoCodeRover+GPT-4* delivered the highest resolution rate of 3.83% among all models. Although relatively costly on average, especially for larger datasets with an average cost of $0.46 per instance, *AutoCodeRover's* effectiveness-aware cost was relatively low at $12.61 per issue fixed, given its high-resolution rate compared to the *SWE-Agent* approach. Meanwhile, *RAG+GPT-3.5* had the lowest average cost, at $0.05 per instance, but a relatively high effectiveness-aware cost of $32 due to its poorer performance in resolving issues correctly.

We recommend that future research not only assess the accuracy of the proposed models but also take into account the financial cost associated with their implementation and operation, ensuring they are not only performant but also practical for large-scale and long-term use.

## 6 RELATED WORK

**LLM for Software Engineering.** Large Language Models (LLMs) have emerged as powerful tools and demonstrated impressive capabilities in various software engineering tasks, including code generation Jiang et al. (2024); Li & Döhmen (2024); Chen et al. (2021); Luo et al. (2024); Du et al. (2024), program repair Zhang et al. (2024b); Yang et al. (2024a); de Fitero-Dominguez et al. (2024) and bug detection Alrashedy & Binjahlan (2024); Hossain et al. (2024). The development of code generation benchmarks has been crucial for evaluating LLM performance. Notably, HumanEval Chen et al. (2021) was introduced to assess the functional correctness of code generated by LLMs. Building on this foundation, AlphaCode Li et al. (2022) demonstrated competitive performance in solving complex programming problems. To address limitations in existing benchmarks, EvalPlus Liu et al. (2024) enhanced HumanEval with more comprehensive test cases and revealed a significant overestimation of LLM performance in previous evaluations. LLMs also have shown promising results in program repair and bug detection. For example, AlphaRepair Xia & Zhang (2022) employed a zero-shot learning approach that outperformed traditional automated program repair (APR) tools. Further research demonstrated that LLMs could surpass existing APR techniques, particularly when fine-tuned on domain-specific data Xia et al. (2023). The application of LLMs in bug detection with innovative approaches like FUzzGPT Deng et al. (2023b) and TitanFuzz Deng et al. (2023a) leveraging these models to generate edge-case test inputs and perform mutation-based fuzzing for deep learning libraries. There are several comprehensive studies have explored LLM applications across various software engineering domains Fan et al. (2023); Hou et al. (2024), delved into the natural language to code generation Zan et al. (2023), and analyzed the evolution and performance of Code LLMs across different tasks Zheng et al. (2024).

**Benchmark Dataset Quality for Code Generation.** To achieve accurate and reliable outcomes in code generation tasks with LLMs, it is essential to use high-quality evaluation benchmark datasets during the training and evaluation steps Jimenez et al. (2024). With the growing interest and significance of code generation and program repair in software engineering, this highlights the critical need for trustworthy and reliable evaluation benchmark datasets Jimenez et al. (2024); Zan et al. (2024). To tackle this challenge, SWE-bench was developed to assist developers in evaluating LLMs using real-world GitHub issues Jimenez et al. (2024). A critical part of making a reliable dataset

is to make it representative of real-world and complex software challenges and issues Chen et al. (2024); Tao et al. (2024). The "Diversity Empowers Intelligence" (DEI) framework Zhang et al. (2024a) further shows how diversity and comprehensiveness of data can enhance the performance of Large Language Models (LLMs) in code generation tasks. Given the critical need for high-quality benchmark datasets in evaluating code generation models, Liu et al. Liu et al. (2024) introduce EvalPlus, a framework specifically designed to improve the evaluation of LLM-generated code. This framework addresses challenges like insufficient tests and noisy benchmarks by augmenting existing datasets with automated test input generation using LLM and mutation-based strategies. Their findings show the need for more rigorous testing in LLM code generation to improve benchmark quality in the field.

## 7 CONCLUSION

In this paper, we presented the first empirical study on the robustness of the *SWE-Bench* dataset. Our study identified significant limitations in the original *SWE-Bench* dataset, particularly issues with solution leakage and weak test cases, which undermined the reliability of previous model assessments. To address these challenges, we introduced SWE-Bench+, a dataset free from solution leakage and built with issues created after LLM training cut-off dates to ensure more rigorous and accurate evaluations. Through extensive testing, we demonstrated that while *SWE-Bench+* resolves the data leakage concerns, weak test cases continue to pose challenges, with model resolution rates dropping further in this refined environment. Despite the reduced pass rates, *SWE-Bench+* establishes a more reliable framework for assessing the true capabilities of LLMs in software development, offering insights into how these models can be better developed and evaluated. Future work should focus on improving the test case robustness in SWE-Bench+ and exploring more effective strategies for filtering data to further minimize biases and inaccuracies.

For future work, further investigation into the issue of weak test cases is needed, along with suggestions for improving test quality to create more accurate and relevant test suites. Another potential avenue of research could explore the underlying causes of the high failure rates and propose strategies to mitigate them. Additionally, similar studies could be conducted on other leading evaluation benchmarks, such as Human-Eval, to compare results and identify broader patterns.

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
