# OpenReview forum: "SWE-Bench+: Enhanced Coding Benchmark for LLMs"
_ICLR.cc/2025/Conference — Submitted to ICLR 2025_

### Official Review · Reviewer_6ZYd · 2024-10-31

**Soundness:** 3
**Presentation:** 2
**Contribution:** 2
**Rating:** 3
**Confidence:** 4

**Summary:**

In this paper, the authors propose SWE-Bench+ to evaluate LLMs in software engineering, specifically for code generation. The authors first reveal that the original SWE-Bench dataset exists solution leakage and weak test cases, which undermine the reliability of the benchmark. The authors propose SWE-Bench+ to address these limitations by collecting GitHub issues created after the LLMs' training cutoff dates and ensuring no solutions are included in the issue reports. The new dataset reduced the pass rate of top-performing LLMs significantly, indicating a more rigorous evaluation.

**Strengths:**

1. The authors reveal the important problems that exist in SWE-Bench, i.e., solution leakage and weak test cases, which are critical for the LLM community.


2. To address these issues, the authors introduce SWE-Bench+, mitigating data leakage by using issues posted after the LLM cutoff dates.

**Weaknesses:**

The primary concerns is from the limited scope of SWE-Bench+. SWE-Bench+ lacks broader comparisons with other LLM code generation benchmarks, and its approach to addressing weak test cases remains limited. Although SWE-Bench+ aims to reduce data leakage and includes more robust cases, additional strategies to enhance test coverage could further improve the benchmark’s reliability. The authors might consider calculating code line and branch coverage of the gold standard code for the provided test cases to ensure their adequacy for experiments.

**Questions:**

1. How does the performance of LLMs on SWE-Bench+ compare with their performance on other common benchmarks like HumanEval or EvalPlus?

2. Given the weak test case issue persists, do the authors have suggestions for specific strategies or criteria to improve test case robustness in SWE-Bench+?

3. Could SWE-Bench+ be expanded to include a wider range of issues beyond GitHub, or is there a specific rationale for its current scope?

4. Can you provide the code line/branch coverage of the test suites in gold solution?

Note:

Could you please check whether this paper has modified the LaTeX style file to create additional space in the main body?

---

### Official Review · Reviewer_wSBt · 2024-11-05

**Soundness:** 2
**Presentation:** 3
**Contribution:** 2
**Rating:** 3
**Confidence:** 4

**Summary:**

This paper conducted an empirical study to analyze the resolved issues by LLMs in the SWE-bench dataset and conclude that the existing benchmark leaks solution inside the issue report, and the test cases are weak. To address this problem, this paper proposed a new dataset SWE-bench+, on which all models have a significant performance drop in terms of resolution rate.

**Strengths:**

- Comprehensive empirical study
- Qualitative examples provided
- Writing is clear and easy to follow

**Weaknesses:**

- Good problem but limited solution

I thank the authors for submitting the work to ICLR. This paper targets a significant topic, i.e., over-optimistic results by LLM for solving an issue. The empirical results are convincing, the paper is easy to follow, and the methodology is well accepted. However, my concern lies in that the important topic has limited solutions to address.

1. Weak test problem

While I agree that the test is weak in SWE-bench, however, I do not see how SWE-bench+ can have strong tests. I would appreciate it a lot if the authors can have a definition of strength and elaborate how SWE-bench+ are equipped with stronger test suites.

2. Data leaksage problem

More importantly, SWT-bench+ can go obsolete when GPT evolves. It indicates that we might soon need a SWT-bench++ in one year, or SWT-bench+++ in three years. With no offense, in this regard, the work can just mitigate the problem temporarily. Thus, what is the silver bullet to address the problem, at least without such considerable efforts for a new benchmark?

**Questions:**

1. How does SWE-bench+ make sure that instances have strong test cases?
2. If GPT-4o further evolve, how does SWE-bench+ evolve accordingly in an automatic way?

---

### Official Review · Reviewer_WuNX · 2024-11-08

**Soundness:** 4
**Presentation:** 3
**Contribution:** 3
**Rating:** 6
**Confidence:** 4

**Summary:**

The paper introduces SWE-Bench+, an enhanced benchmark to assess the real-world problem-solving capabilities of LLMs in software engineering. This work follows up on the original SWE-Bench and its variants, identifying significant shortcomings, such as solution leakage and weak test cases. The authors highlight how these issues affect the perceived effectiveness of LLMs and propose SWE-Bench+ to mitigate these limitations by excluding solution leaks and using issues created after LLMs' cutoff dates.

**Strengths:**

- Comprehensive Analysis: The paper provides a thorough empirical analysis of the original SWE-Bench, clearly explaining issues like solution leakage and insufficient test case robustness.
- Clear Motivation: The rationale behind SWE-Bench+ is well-articulated, emphasizing a more realistic evaluation framework by excluding solution leaks and data leakage that could bias LLM performance.
- Empirical Evidence: The study offers strong quantitative results, demonstrating how stricter criteria drastically lower LLM resolution rates, underscoring the need for more robust benchmarks.

**Weaknesses:**

- Test Case Quality: Although SWE-Bench+ addresses solution leakage, the issue of weak test cases persists. Future work could provide more advanced strategies for enhancing test case robustness.
- Reliance on Manual Effort: The exclusion of solution leaks in SWE-Bench+ heavily depends on human labeling, which poses scalability challenges. This manual methodology may not be feasible for extending the approach to larger or more comprehensive benchmarks.

**Questions:**

1. Are there any ongoing efforts to incorporate automated or community-sourced improvements for strengthening test cases in SWE-Bench+?
2. Is it possible to leverage advanced LLMs to automatically identify and exclude solution leaks and low-quality instances?

---

### Official Review · Reviewer_JY4G · 2024-11-23

**Soundness:** 3
**Presentation:** 2
**Contribution:** 2
**Rating:** 3
**Confidence:** 4

**Summary:**

This paper analyzes the generation results of SWE-Agent+GPT-4 on SWE-bench, highlighting certain shortcomings within the original SWE-bench. For example, the authors found that nearly one-third of instances in the dataset contain the answer directly within the issues, preventing a fair and accurate assessment of the model’s ability to solve such problems. Additionally, there are cases where the model modifies parts that should remain unchanged or makes incorrect changes that still pass incomplete test cases. Based on these issues, the authors propose an improved benchmark, SWE-bench+, which filters out issues containing the correct answer. Testing on this benchmark revealed that, under these conditions, the model struggles to accurately solve these types of problems.

**Strengths:**

The motivation of this paper is clear. The study of SWE-bench is thorough, with detailed classification and discussion of various imprecise answers and data, providing valuable insights.

**Weaknesses:**

The **SWE-BENCH+** section in this paper has certain limitations in terms of contribution, presentation, and the adequacy of experiments:

- **Presentation**: The section lacks a basic introduction to the model methods, as well as a detailed analysis of results and experimental configuration settings. More importantly, presenting model experimental results by listing them in paragraph form is unclear; a table format would enhance clarity.

- **Contribution**: SWE-BENCH+ appears to differ from the original SWE-bench only by filtering out issues that contain answers and setting a newer cutoff date. Beyond these changes, there are no additional distinctions, and among the three issues highlighted in the empirical study, only the "direct copy of the answer" issue is addressed in this new benchmark. The problem of incomplete test cases remains unresolved, casting doubt on SWE-BENCH+ as a sufficiently rigorous evaluation dataset, as claimed. Given this, I find the benchmark’s contribution limited.

- **Experiments**: For a newly proposed benchmark, the experiments only assess model performance and cost, which is insufficient. More comprehensive analyses should have been conducted.

**Questions:**

-  Why not use a table or diagram to present your experimental results? Listing them in paragraph form makes it hard to interpret the data clearly.

- I’m also curious about the example in Figure 4. How does the model pass the test case even with an incorrect implementation?

---

### Meta-Review · Area_Chair_kdTa · 2024-12-21

**Metareview:**

This paper presents SWE-Bench+, an extened evaluation of the original SWE-Bench dataset, to evaluate the real-world problem-solving capabilities of LLMs in software engineering. While valuable, the reviewers have raised key concerns. Firstly, SWE-Bench+ lacks broader comparisons with other established LLM code generation benchmarks, which limits its contextual relevance. Moreover, the reviewers note that SWE-Bench+ risks becoming obsolete as models like GPT continue to evolve rapidly, questioning the necessity of this specific evaluation framework. As such, a systematic technical evaluation/methdology is required, which can shine beyond the evolution of GPT family. The authors are encouraged to address these issues, particularly by broadening comparisons and strengthening the motivation for SWE-Bench+, and to consider resubmitting after significant revisions.

**Additional Comments On Reviewer Discussion:**

Several key issues were identified during the discussions; while some have been addressed, others remain insufficiently explored.

---

### Decision · Program_Chairs · 2025-01-22

Reject